# Utilization of Corn Steep Liquor for the Production of Fairy Chemicals by *Lepista sordida* Mycelia

**DOI:** 10.3390/jof8121269

**Published:** 2022-12-01

**Authors:** Hajime Kobori, Jing Wu, Hirohide Takemura, Jae-Hoon Choi, Naoto Tada, Hirokazu Kawagishi

**Affiliations:** 1Iwade Research Institute of Mycology Co., Ltd., 1-9 Suehiro, Tsu 514-0012, Japan; 2Research Institute for Mushroom Science, Shizuoka University, 836 Ohya, Suruga-ku, Shizuoka 422-8529, Japan; 3Faculty of Agriculture, Shizuoka University, 836 Ohya, Suruga-ku, Shizuoka 422-8529, Japan; 4Graduate School of Science and Technology, Shizuoka University, 836 Ohya, Suruga-ku, Shizuoka 422-8529, Japan; 5Research Institute of Green Science and Technology, Shizuoka University, 836 Ohya, Suruga-ku, Shizuoka 422-8529, Japan

**Keywords:** *Lepista sordida*, cultured mycelia, 2-azahypoxanthine, imidazole-4-carboxamide, corn steep liquor

## Abstract

There are various potential practical uses of fairy chemicals (FCs) in the fields of agriculture, cosmetics, and medicine; however, the production costs of FCs are very high. To enable the practical use of FCs, more efficient and inexpensive methods of culturing the mycelia of FCs-producing fungi and producing FCs need to be developed. The purpose of the present study was to determine methods of reducing the production costs of FCs and mycelia of the FCs-producing fungus *Lepista sordida*. We investigated the effects of four food industrial by-products, i.e., corn steep liquor (CSL), rice bran, wheat bran, and Japanese liquor lees, as nutritional additives in the liquid culture medium of the fungus. We found that CSL was more effective than the other tested additives in increasing the production of FCs and mycelia. Medium containing 1% CSL was optimal for increasing the mycelial yield while medium containing 6% CSL was optimal for increasing the production of FCs. The reason for this difference in the optimal CSL concentration was considered to be related to the stress on the mycelia caused by the amount of nutrients in the liquid medium. These results are expected to facilitate the practical use of FCs and the mycelia of FCs-producing fungi.

## 1. Introduction

Rice, wheat, and corn are the most abundantly produced grains in the world [1]. In the processing of these grains for foods, corn steep liquor (CSL), rice bran, wheat bran, and Japanese liquor lees are generated as food industrial by-products. Since most of these are discarded as wastes, the development of new usages for these by-products is desired. While rice bran and wheat bran are commonly used as nutritional additives in the artificial cultivation of mushrooms [2], CSL and Japanese liquor lees are rarely used. In addition, these four by-products are rarely used as nutritional additives in liquid culture media for the mycelial cultivation of mushroom-forming fungi. However, the production costs of mycelia as well as useful substances from the mycelia may be reduced by optimizing the liquid medium used for mycelial cultivation. In fact, it has been reported that the mycelial yield of *Grifola gargal,* and the production of characteristic components of the fungus, benzaldehyde and ergothioneine, were increased by using the food industrial by-products from the distillery waste of a buckwheat alcohol beverage company as a nutritional additive in liquid culture when compared the use of medium containing only reagents, such as yeast extract and polypeptone [3].

In previous studies, we identified the compounds responsible for the fairy ring phenomenon, and we named them fairy chemicals (FCs) [4]. FCs consist of three compounds: 2-azahypoxanthine (AHX), imidazole-4-carboxamide (ICA), and 2-aza-8-oxohypoxanthine (AOH). AHX and ICA produced by *Lepista sordida*, a fairy ring-forming fungi, were involved in the growth and death of turfgrass, respectively [5,6]. Although AHX and ICA have been chemically synthesized [7,8,9,10], they were first isolated from a natural source. AHX has plant growth-promoting activity while ICA has growth-suppressing activity against various plants [5,6]. After the discovery of AHX and ICA, a novel compound, AOH, was discovered as a metabolite of AHX in plants, and it showed similar activity as AHX [11]. FCs are contained in all the plants tested regardless of the taxonomic families [11,12,13], and various studies have been conducted under the hypothesis that FCs are a novel family of plant hormones. In addition, we elucidated the molecular mechanism of the plant growth-regulating activity of FCs and the biosynthetic routes of AHX and AOH in a novel purine metabolic pathway in plants [12,13]. Since FCs show growth regulatory activity in all of the examined plants thus far, including crops, such as rice and wheat, studies on the molecular mechanism of their activity and field experiments aimed at applying the compounds to agriculture have been performed [12,13,14].

Animal experiments have indicated that AHX suppresses retinal angiogenesis as a hypoxia-inducible factor inhibitor, and that ICA inhibits the expression of Anexelekto receptor tyrosine kinase and immune checkpoint molecules (programmed death-ligand 1 and 2), and improves the response to cisplatin in cancer [15,16]. As an example of the commercial use of FCs, we found that AOH had beneficial effects on human skin [17,18,19]; we proved its safety by performing various tests, including human studies, and as a result, AOH will be launched as a cosmetic ingredient in 2022 [20,21]. However, AOH is currently expensive as a raw material for cosmetics, and methods of reducing the production costs are desired.

We are currently examining practical uses of FCs in agriculture and medicine. AOH is prepared from AHX at a relatively low cost by microbial conversion [22]. However, the production costs of AHX and ICA are very high at the present time, because they are prepared by chemical synthesis from the precursor compound 5-aminoimidazole-4-carboxamide (AICA), which has little circulation in the general market and is expensive. Although we tried to find less costly methods of synthesizing FCs and AICA, no suitable method has been established yet. This high production cost remains the largest obstacle for the practical use and adoption of FCs in agriculture, cosmetics, and medicine. Therefore, in the present study, we determined the optimal conditions for culturing the mycelia of *L. sordida* and producing AHX and ICA using food industrial by-products as additives in liquid culture medium.

## 2. Materials and Methods

### 2.1. Fungal Material

Mycelia of *L. sordida* (Schuml: Fr.) Sing. (NBRC No. 112841) was obtained from the Biological Resource Center, National Institute of Technology and Evaluation (NBRC; Kisarazu, Japan).

### 2.2. Compositions of the Media

Four food industrial by-products, CSL (Oji Cornstarch Co., Ltd., Tokyo, Japan), rice bran (Rice Shop Imai, Tsu, Japan), wheat bran (Japan Agricultural Cooperatives Matsusaka, Matsusaka, Japan), and Japanese liquor lees (Ota Sake Brewery Co., Ltd., Iga, Japan), were used in the liquid culture media (Table 1, Table 2, Table 3 and Table 4). Yeast and glucose (YG) medium was used as the control medium. The compositions of the media are shown in Table 1. Nineteen media were used in the experiments; (1) CSL, (2) Rice bran, (3) Wheat bran, (4) Japanese liquor lees, (5) CSL-1% (Salts-0.1 + 0.1% Glc-4% CSL-1%), (6) CSL-2%, (7) CSL-3%, (8) CSL-6% (Salts-0.1 + 0.1% Glc-4% CSL-6%), (9) Salts-0.1 + 0.1% Glc-2% CSL-1%, (10) Salts-0.1 + 0.1% Glc-1% CSL-1%, (11) Salts-0% Glc-4% CSL-1%, (12) Salts-0% Glc-2% CSL-1%, (13) Salts-0% Glc-1% CSL-1%, (14) Salts-0.1 + 0.1% Glc-2% CSL-6%, (15) Salts-0.1 + 0.1% Glc-1% CSL-6%, (16) Salts-0% Glc-4% CSL-6%, (17) Salts-0% Glc-2% CSL-6%, (18) Salts-0% Glc-1% CSL-6% and (19) control (YG). Each reagent and nutritional additive was sequentially dissolved in water with a stirrer, and 1 N hydrochloric acid (for media containing rice bran, wheat bran, or Japanese liquor lees, and the control medium) or 1 N aqueous sodium hydroxide (for media containing CSL) was added until the pH reached 5.5 ± 0.1. The liquid media (100 mL each) were dispensed into separate 500-mL Erlenmeyer flasks, and sterilized by autoclaving at 121 °C and 1.2 atm for 20 min. After sterilization, all of the flasks were placed on a clean bench overnight to cool to room temperature.

### 2.3. Culture Conditions

The purchased mycelia of *L. sordida* arrived in a slant medium. The mycelia were placed onto the center of a Petri dish containing potato dextrose agar (PDA), and incubated in the dark at 25 °C for 7 days in an incubator (Advantec Incubator CI-610; Advantec Toyo Kaisha, Ltd., Tokyo, Japan). After incubation, the PDA medium was covered with mycelia, and a block was extracted with a cork borer and used for inoculating autoclaved YG medium (100 mL in a 500 mL Erlenmeyer flask) aseptically. The inoculated medium was cultured in the dark at 25 °C with shaking (100 rpm) for 14 days in an incubator (MW-316; ABLE Co. Ltd., Tokyo, Japan). After the incubation, the culture was homogenized with a sterilized homogenizer (Cell Master CM-100; AS ONE Co., Ltd., Osaka, Japan), and 5 mL of the homogenized culture was inoculated into each of the test media or control medium (100 mL/500 mL Erlenmeyer flask) aseptically. The inoculated media were cultured in the dark at 25 °C with shaking (100 rpm) for 14 days. After the cultivation, each culture was filtered to separate the mycelia and culture filtrate. The mycelia were dried in a warm air drier (DRM420DB Electric Drying Oven; Advantec Toyo Kaisha Ltd.) at 50 °C for 14 h, then at 70 °C for 2 h. The resulting dried mycelia were weighed.

### 2.4. Quantification of the FCs

The FCs were quantified as previously described [23]. The culture filtrates were diluted by a factor of 10,000 with 80% acetonitrile, and subjected to liquid chromatography-mass spectrometry (LC-MS/MS) analysis. Allopurinol was used as an internal standard.

The quantification of FCs was performed using a Waters ACQUITY H-class UPLC system (Waters; Milford, MA, United States) connected to a Waters Xevo TQ-S micro mass spectrometer equipped with an electrospray ionization probe (Waters). MS/MS spectra were detected by a tandem quadrupole mass spectrometer in the positive ion mode. A PC-HILIC column (diameter of 2 × 100 mm, 3 μm; Shiseido, Tokyo, Japan) was used in the analysis (injection volume: 2 μL; solvent: 95% acetonitrile with 0.05% formic acid; and flow rate: 0.4 mL/min). The ion source conditions were as follows: capillary voltage: 3.50 kV; cone voltage: 30 V; desolvation temperature: 600 °C; desolvation gas flow: 1000 L/h; and cone gas flow: 50 L/h. The compounds were identified by the characteristic transitions at *m/z* 138 > 67 for AHX (cone voltage: 26 V; collision energy: 16 V), *m/z* 112 > 95 for ICA (cone voltage: 15 V; collision energy: 11 V), and *m/z* 137 > 110 for allopurinol (cone voltage: 26 V; collision energy: 18 V). Data acquisition and analysis were performed using MassLinx software (Waters).

### 2.5. Statistical Analysis

One-way analysis of variance and the Tukey–Kramer post hoc test were used for data analysis. All analyses were performed using R software (https://www.r-project.org/ (accessed on 1 October 2022)).

## 3. Results

### 3.1. Effects of Food Industrial By-Products as Nutritional Additives on the Mycelial Cultivation of L. sordida

After shaking cultivation in each liquid medium for 14 days, the medium containing CSL (CSL medium) gave the best results; the dry weight of the mycelia and FCs contents in the medium were similar to those in the positive control, respectively (Figure 1 and Figure 2). There is a report that concentration of CSL changes production amount of ethanol by yeast and *candida shehatae* [24]. This previous report and the result in this study indicated that low-cost CSL is much better than control (YG). Therefore, effect of concentrations of CSL was examined.

### 3.2. Investigation of the Optimal Concentration of CSL as a Nutritional Additive for the Mycelial Cultivation of L. sordida

To determine the optimal amount of CSL in the medium for the cultivation of *L. sordida* mycelia, liquid media supplemented with 1% to 6% CSL (CSL-1% to -6% media) were used to cultivate *L. sordida* mycelia for 14 days (Table 2). The dry weight of the cultivated mycelia was the highest in CSL-1% medium, and was 1.77 times that of the control (Figure 3). The FCs contents were the highest in the CSL-6% medium; the AHX content was 8.58 times that of the control, and the ICA content was 3.79 times that of the control (Figure 4). Concentrations of CSL higher than 6% strongly inhibited mycelial growth. These results indicated that the optimal concentration of CSL is 6% or less.

### 3.3. Effects of Glucose and Inorganic Salts on the Mycelial Cultivation of L. sordida

As shown in Figure 3 and Figure 4, the optimal medium for mycelial growth (CSL-1% medium) and that for FCs production (CSL-6% medium) were different. To further reduce the cost of the medium, *L. sordida* was cultured in CSL-1% and -6% media with reduced concentrations of glucose and inorganic salts (Table 3 and Table 4). Media containing CSL-1% or -6%, 1%, 2% or 4% glucose (Glc-1%, -2%, or -4%, respectively), and 0% or 0.1 + 0.1% inorganic salts (Salts-0% or -0.1 + 0.1%, respectively) were used. After cultivation with shaking in each liquid medium for 14 days, the mycelial yields and ICA content were reduced in all of the tested media when compared to the Salts-0.1 + 0.1% Glc-4% CSL-1% medium. However, the AHX content was increased in the Salts-0.1 + 0.1% Glc-2% CSL-1% medium, Salts-0.1 + 0.1% Glc-1% CSL-1% medium, and Salts-0% Glc-1% CSL-1% medium (Figure 5 and Figure 6). The decrease in the mycelial yield was lower in the Salts-0.1 + 0.1% Glc-2% CSL-6% medium than in the CSL-6% (Salts-0.1 + 0.1% Glc-4% CSL-6%) medium (Figure 7). On the other hand, the FCs contents were decreased in all of the media, except for the AHX content in the Salts-0.1 + 0.1% Glc-2% CSL-6% medium (Figure 8).

## 4. Discussion

### 4.1. CSL as a Nutritional Additive in Liquid Media Is Effective for the Culturing of L. sordida Mycelia

In this study, we determined the optimal conditions for culturing *L. sordida* mycelia using CSL, which is a food industry by-product from the manufacturing process of cornstarch. Approximately 150,000 tons of CSL are generated annually in Japan, and although some of it is used in microbial media and livestock feed, new applications of CSL are needed. We found that CSL-1% medium and CSL-6% medium were optimal for increasing the mycelial yield and the FCs contents, respectively, when compared to YG medium, which was used for liquid cultures in previous studies (Figure 3 and Figure 4). CSL has been used for liquid cultures of bacteria in the industrial production of bioactive compounds, such as penicillin and riboflavin [25,26]. In addition, it has been reported that the use of glucose and CSL enabled the production of succinic acid by *Anaerobiospirillum succiniciproducens* to become economically feasible [27]. Furthermore, the effects of CSL supplementation on fruiting body formation have been reported in the sawdust-based cultivation of *Lentinula edodes* [28]. However, CSL has rarely been used for the mycelial cultivation of mushroom-forming fungi to produce bioactive compounds in liquid culture. Yeast extracts in YG medium are much more expensive than CSL [29]. The results of the present study indicate that CSL medium can be used for the efficient and inexpensive cultivation of FCs-producing fungi and the production of AHX and ICA.

### 4.2. Speculation as to Why CSL Increased the Mycelial Yield, and AHX and ICA Contents

CSL contains inorganic substances (potassium, phosphorus, and magnesium, etc.), vitamins (inositol, choline, and niacin, etc.), saccharides (glucose, fructose, and galactose, etc.), and amino acids (glutamic acid, proline, and alanine, etc.) that are necessary for mycelial growth [28,30]. In addition, zeanic acid (2,8-dihydroxycinchoninic acid) has been isolated as a plant growth promoter from CSL [31,32]. These factors may have been responsible for the increased mycelial growth, and AHX and ICA contents in the present study. It is noteworthy, however, that AHX and ICA production was the highest in the CSL-6% medium, which was the medium that produced the least mycelia. In particular, in the CSL-1% medium with a reduced amount of glucose, the AHX content increased while the mycelial yield decreased (Figure 5 and Figure 6). In contrast, in the CSL-6% medium with a reduced amount of glucose, the AHX and ICA contents decreased while the mycelial yield increased (Figure 7 and Figure 8). The concentration of inorganic salts significantly affected the mycelial yield, and the AHX and ICA contents; CSL-1% medium with 0.1% MgSO_4_·7H_2_O and 0.1% KH_2_PO_4_ resulted in a significantly increased mycelial yield when compared to CSL-1% medium with no salts, and CSL-6% medium with the salts resulted in significantly increased AHX and ICA contents when compared to CSL-6% medium with no salts. On the other hand, with the Salts-0.1 + 0.1% Glc-1% CSL-1% medium and Salts-0% Glc-1% CSL-1% medium, the presence or absence of the inorganic salts did not affect AHX production. These results indicated that inorganic salts are not essential for increasing AHX production, but are important for increasing ICA production and the mycelial yield. These results also suggested that the stress on *L. sordida* caused by an excess or deficiency of nutrients in the liquid media reduced the mycelial yield, but increased AHX production. Studies on the mechanism of AHX activity have revealed that rice acquires resistance to the continuous stress from diverse environmental factors through AHX [13]. Therefore, it is speculated that the production of AHX was increased in *L. sordida* to gain tolerance to the stress conditions. Plants produce various defensive low molecule compounds, such as phytoalexins and allelopathic compounds, in response to external stress [33,34,35,36,37,38]. However, these kinds of compounds in mushroom-forming fungi remain almost unknown. Previously, we showed that AHX exists endogenously not only in plants, but also in fungi, and that it is biosynthesized through a novel purine metabolic pathway [12,13]. Furthermore, the ubiquitous presence of AHX in mushroom-forming fungi, such as *Cortinarius caperatus*, *Flammulina velutipes*, *Grifola frondosa*, *Hypholoma sublateritium*, *Lepista nuda*, *Lyophyllum connatum*, *Lyophyllum shimeji*, *Lyophyllum decastes*, *Pholiota adiposa,* and *Tricholoma flavovirens,* regardless of the taxonomic family, indicated that AHX might act as a stress-tolerant compound in fungi as well [12,13,39].

### 4.3. Efficient Mycelial Cultivation Method for L. sordida According to the Purpose

The results of the present study may contribute to the establishment of an efficient method for the mycelial cultivation of *L. sordida* on a plant scale in the future. Since the culture conditions for obtaining a high mycelial yield (CSL-1% medium) and high AHX and ICA contents (CSL-6% medium) differed, the appropriate concentration of CSL to use should be chosen depending on the purpose of the culture. For example, when a large amount of mycelial extract or crushed mycelia is needed, the CSL-1% medium should be selected. In contrast, when a large amount of AHX and ICA is needed, the CSL-6% medium should be selected. Since the bioconversion reaction from AHX to AOH can be performed at a relatively low cost [22], it is possible to produce any of the FCs (AHX, ICA, and AOH) in large quantities from the mycelia of FCs-producing fungi as they can be easily and inexpensively produced using the optimized conditions reported in the present study. Therefore, the mycelia and the culture broth of *L. sordida* cultured in CSL medium may be applicable as a cost-effective and more active fertilizer, because the synergic or additive effect of CSL, AHX and ICA [40,41,42].

## 5. Conclusions

In the present study, we determined efficient and inexpensive methods for obtaining FCs and mycelia from an FCs-producing fungi, *L. sordida,* using CSL, a food industrial by-product. Specifically, CSL-1% medium (Table 1 and Table 3) was effective for obtaining a high mycelial yield, and CSL-6% medium (Table 2 and Table 4) was effective for obtaining high FCs production. CSL is a highly concentrated acidic organic by-product that is difficult to reuse due to high costs. However, discharging CSL in wastewater is not only a waste of resources, but also a major cause of environmental pollution. The development of practical uses for CSL is a major issue faced by companies and society [43]. Our results are expected to contribute to society by reducing wastes and protecting the environment through the practical use of CSL for the commercial production of FCs and mycelia from FCs-producing fungi.

## Figures and Tables

**Figure 1 jof-08-01269-f001:**
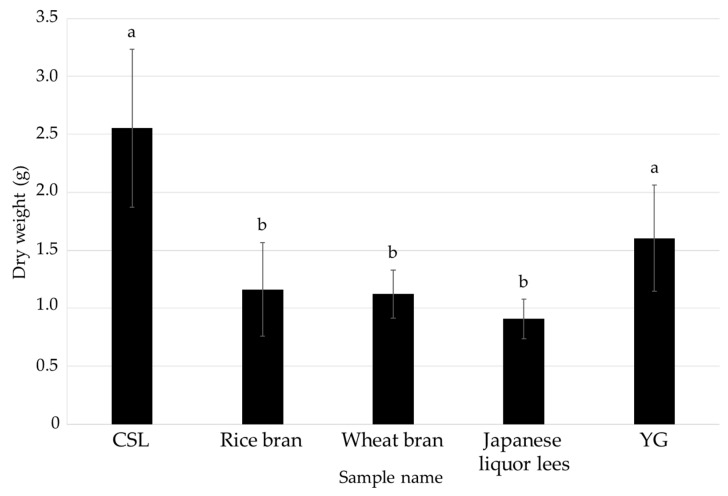
Effects of the media containing each of the food industrial by-products on the mycelial growth of *L. sordida*. The dry weight of the mycelia (the water-insoluble components) for each condition is shown. Different alphabet letters indicate significant differences (Tukey–Kramer post hoc test, *p* < 0.05; *n* = 3).

**Figure 2 jof-08-01269-f002:**
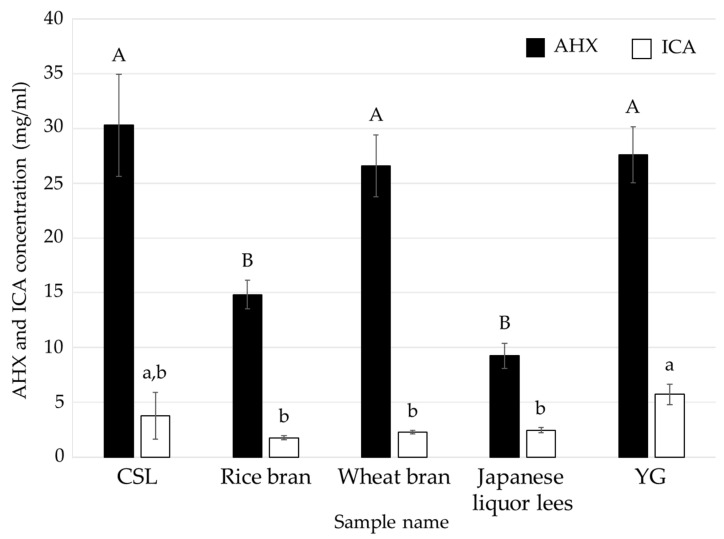
AHX and ICA contents of the culture filtrates of *L. sordida* mycelia cultured in media containing a food industrial by-product or YG. Different alphabet letters (capital letters for AHX, and small letters for ICA) indicate significant differences (Tukey–Kramer post hoc test, *p* < 0.05; *n* = 3).

**Figure 3 jof-08-01269-f003:**
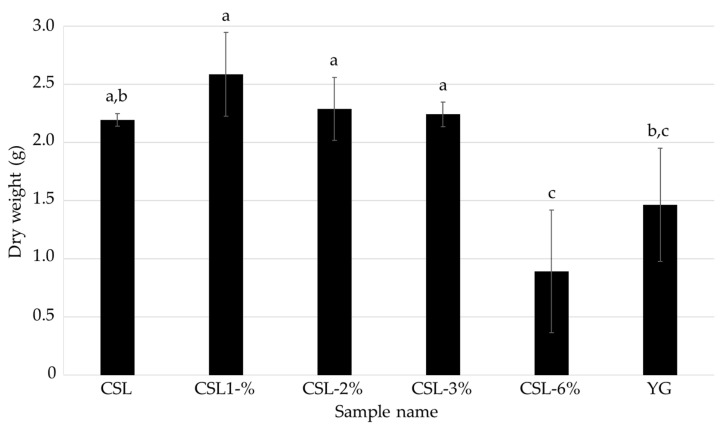
Effects of the media containing various concentrations of CSL on the mycelial growth of *L. sordida*. Different alphabet letters indicate significant differences (Tukey–Kramer post hoc test, *p* < 0.05; *n* = 4).

**Figure 4 jof-08-01269-f004:**
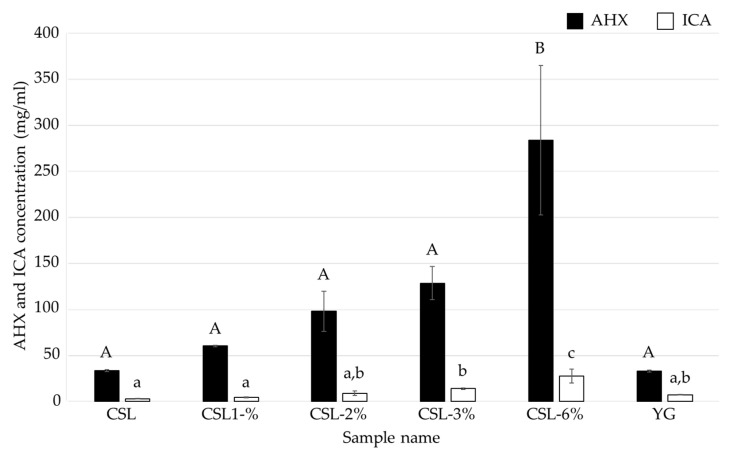
AHX and ICA contents of the culture filtrates of *L. sordida* mycelia cultured in media containing various concentrations of CSL. Different alphabet letters (capital letters for AHX, and small letters for ICA) indicate significant differences (Tukey–Kramer post hoc test, *p* < 0.05; *n* = 3).

**Figure 5 jof-08-01269-f005:**
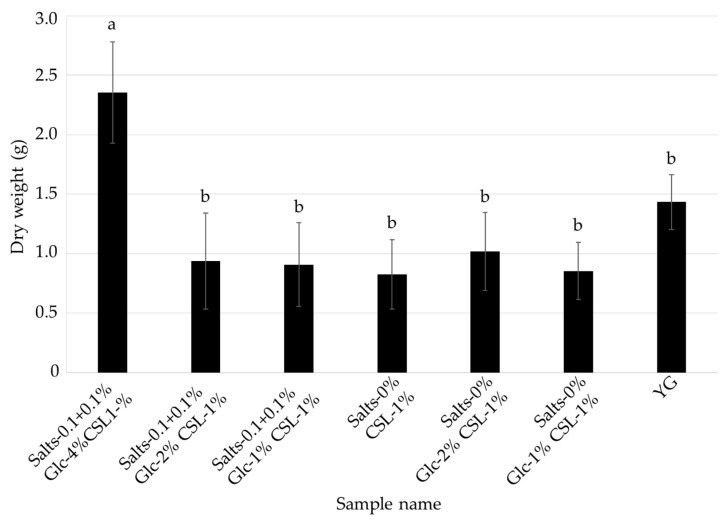
Effects of the CSL-1% media with reduced concentrations of glucose and inorganic salts on the mycelial growth of *L. sordida*. Different alphabet letters indicate significant differences (Tukey–Kramer post hoc test, *p* < 0.05; *n* = 3).

**Figure 6 jof-08-01269-f006:**
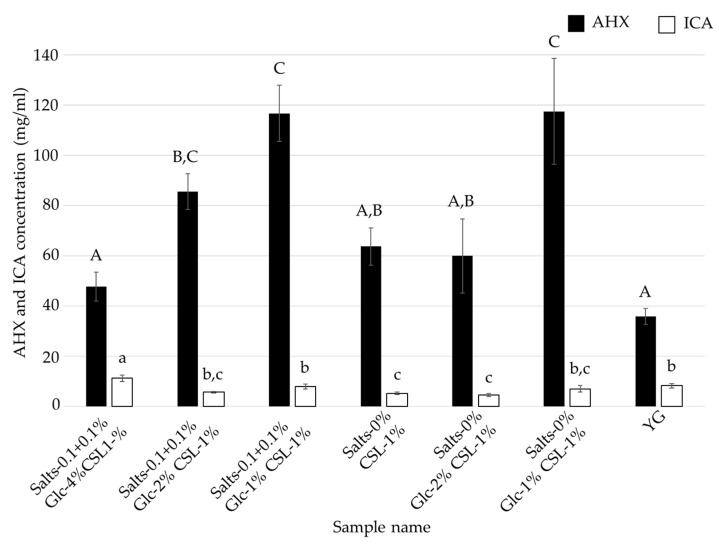
AHX and ICA contents of the culture filtrates of *L. sordida* mycelia cultured in CSL-1% media with reduced concentrations of glucose and inorganic salts. Different alphabet letters (capital letters for AHX, and small letters for ICA) indicate significant differences (Tukey–Kramer post hoc test, *p* < 0.05; *n* = 3).

**Figure 7 jof-08-01269-f007:**
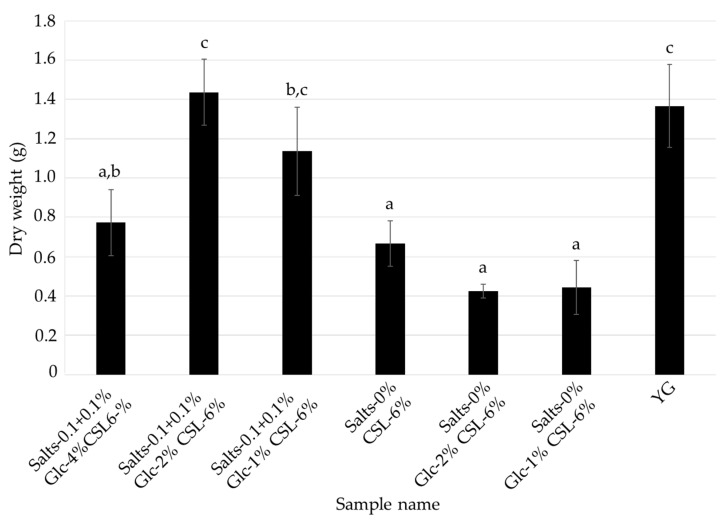
Effects of the CSL-6% media with reduced concentrations of glucose and inorganic salts on the mycelial growth of *L. sordida*. Different alphabet letters indicate significant differences (Tukey–Kramer post hoc test, *p* < 0.05; *n* = 3).

**Figure 8 jof-08-01269-f008:**
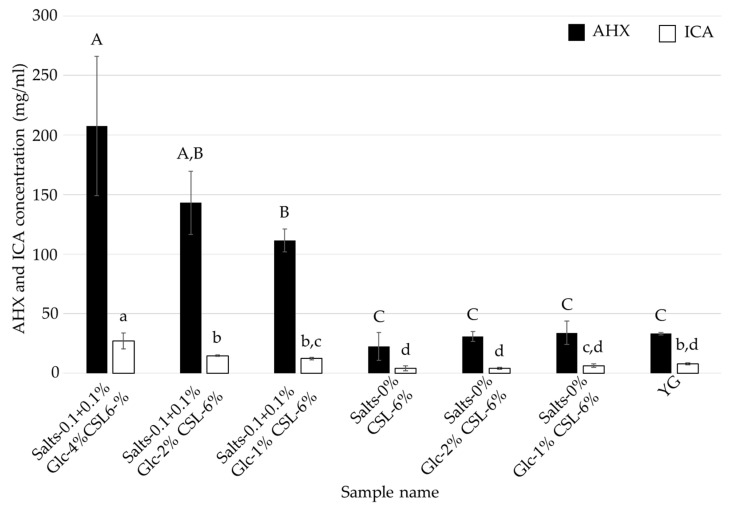
AHX and ICA contents of the culture filtrates of *L. sordida* mycelia cultured in CSL-6% media with reduced concentrations of glucose and inorganic salts. Different alphabet letters (capital letters for AHX, and small letters for ICA) indicate significant differences (Tukey–Kramer post hoc test, *p* < 0.05; *n* = 3).

**Table 1 jof-08-01269-t001:** Compositions of the media containing food industrial by-products, and the control (YG) medium. All media were adjusted to a pH of 5.5 ± 0.1.

	CSL	Rice	Wheat	Japanese	Control
		Bran	Bran	Liquor Lees	(YG)
Glucose	4%	4%	4%	4%	4%
CSL	0.3%	-	-	-	-
Rice bran	-	0.3%	-	-	-
Wheat bran	-	-	0.3%	-	-
Japanese liquor lees	-	-	-	0.3%	-
Yeast extract	-	-	-	-	0.3%
MgSO_4_·7H_2_O	0.1%	0.1%	0.1%	0.1%	0.1%
KH_2_PO_4_	0.1%	0.1%	0.1%	0.1%	0.1%

**Table 2 jof-08-01269-t002:** Compositions of the media containing various concentrations of CSL. All media were adjusted to a pH of 5.5 ± 0.1.

	CSL	CSL-1%	CSL-2%	CSL-3%	CSL-6%	Control
						(YG)
Glucose	4%	4%	4%	4%	4%	4%
CSL	0.3%	1%	2%	3%	6%	-
Yeast extract	-	-	-	-	-	0.3%
MgSO_4_·7H_2_O	0.1%	0.1%	0.1%	0.1%	0.1%	0.1%
KH_2_PO_4_	0.1%	0.1%	0.1%	0.1%	0.1%	0.1%

**Table 3 jof-08-01269-t003:** Compositions of the CSL-1% media with reduced concentrations of glucose and inorganic salts. All media were adjusted to a pH of 5.5 ± 0.1.

	Salts- 0.1 + 0.1% Glc-4% CSL-1%	Salts- 0.1 + 0.1% Glc-2% CSL-1%	Salts- 0.1 + 0.1% Glc-1% CSL-1%	Salts-0% Glc-4% CSL-1%	Salts-0% Glc-2% CSL1%	Salts-0% Glc-1% CSL-1%	Control (YG)
Glucose	4%	2%	1%	4%	2%	1%	4%
CSL	1%	1%	1%	1%	1%	1%	-
Yeast extract	-	-	-	-	-	-	0.3%
MgSO_4_·7H_2_O	0.1%	0.1%	0.1%	-	-	-	0.1%
KH_2_PO_4_	0.1%	0.1%	0.1%	-	-	-	0.1%

**Table 4 jof-08-01269-t004:** Compositions of the CSL-6% media with reduced concentrations of glucose and inorganic salts. All media were adjusted to a pH of 5.5 ± 0.1.

	Salts- 0.1 + 0.1% Glc-4% CSL-6%	Salts- 0.1 + 0.1% Glc-2% CSL-6%	Salts- 0.1 + 0.1% Glc-1% CSL-6%	Salts-0% Glc-4% CSL-6%	Salts-0% Glc-2% CSL-6%	Salts-0% Glc-1% CSL-6%	Control (YG)
Glucose	4%	2%	1%	4%	2%	1%	4%
CSL	6%	6%	6%	6%	6%	6%	-
Yeast extract	-	-	-	-	-	-	0.3%
MgSO_4_·7H_2_O	0.1%	0.1%	0.1%	-	-	-	0.1%
KH_2_PO_4_	0.1%	0.1%	0.1%	-	-	-	0.1%

## Data Availability

All data are available from the corresponding author upon reasonable request.

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
