# Peer review of "Utilization of Corn Steep Liquor for the Production of Fairy Chemicals by Lepista sordida Mycelia"

_jof, 2022, doi:10.3390/jof8121269_

Round 1

Reviewer 1 Report

-The authors work is relevant especially making worth out of corn steep liquor and making it a competitive nutritional source.

-Please check the following and discuss all observations and findings scientifically. Check the attached pdf too.

-Line 19-24: The author should justify how the optimum was carried out?

-line 25-27: -Redefine the keywords

--line72-75-justify the sentence with the appropriate references

-line 88-93: What is the basis for this methodogy? I mean references or your past protocol. IN Hcl or IN NaOH? Why OR, is the author not sure of the method used?

--line 96. reference this please

-line 110-paraphrase

-line 130-131: what did you use for your statistical package or software did you use for this analysis

-Line 144-146-Justify the reason for this scientifically not just by observation. 

Check the following article if it could be of help or any other article already listed in your manuscript references to support all your observations scientifically.

https://doi.org/10.3390/en11071740.

https://link.springer.com/article/10.1007/s12010-022-03904-w

--line 168-172; justify your observation scientifically and with proper reference

-Line 197-200 :Justify the statement with scientific reasons and make reference to literature 

--line 257-264: Justify this statement scientifically

-line 292-295: Justify a scientific reason for this and make proper references

-Line 307-311-I suggest you re-write and update your conclusions. Include benefits of the studies to industries especially in waste management sector. Also recommend a well design optimization studies in the future research

Author Response

Dear Reviewer #1

Thank you for your review about our manuscript entitled “Utilization of corn steep liquor for production of fairy chemicals by Lepista sordida mycelia”. We revised the manuscript according to your comments and another reviewer’s. The English has been improved by a professional service company (see the attached English proofreading certificate). The corrected portions by the service have been highlighted in light-green and the corrected portions according to your suggestion have been highlighted in light-blue.

Your comments

The authors work is relevant especially making worth out of corn steep liquor and making it a competitive nutritional source. Please check the following and discuss all observations and findings scientifically. Check the attached pdf too.

1) Line 19-24: The author should justify how the optimum was carried out?

2) Line 25-27: Redefine the keywords

3) Line72-75: justify the sentence with the appropriate references

4) Line 88-93: What is the basis for this methodogy? I mean references or your past protocol. IN Hcl or IN NaOH? Why OR, is the author not sure of the method used?

5) Line 96: reference this please

6) Line 110: paraphrase

7) Line 130-131: what did you use for your statistical package or software did you use for this analysis

8) Line 144-146: Justify the reason for this scientifically not just by observation.

Check the following article if it could be of help or any other article already listed in your manuscript references to support all your observations scientifically.

https://doi.org/10.3390/en11071740.

https://link.springer.com/article/10.1007/s12010-022-03904-w

9) Line 168-172: justify your observation scientifically and with proper reference

10) Line 197-200: Justify the statement with scientific reasons and make reference to literature

11) Line 257-264: Justify this statement scientifically

12) Line 292-295: Justify a scientific reason for this and make proper references

13) Line 307-311: I suggest you re-write and update your conclusions. Include benefits of the studies to industries especially in waste management sector. Also recommend a well design optimization studies in the future research

Response to your comments

1) This part describes the summary of the experimental results in Figure 1-4. Therefore, the “optimum” was determined according to the experimental results. The sentence has been changed to “Medium containing 1% CSL was optimal for increasing the mycelial yield while medium containing 6% CSL was optimal for increasing the production of FCs” (lines 20-22).

2) Keywords were confined (lines 25-26).

3) The sentence "Although we tried to find less costly methods of synthesizing FCs and AICA, no suitable method has been established yet" has been added (lines 74-76).

4) pH of the CSL medium was around 3.0-4.0, and sodium hydroxide was added to bring the pH to 5.5. The pH of other media was around 5.8-6.1, and were adjusted to pH 5.5 with hydrochloric acid. Therefore, the words "(media containing rice bran, wheat bran or Japanese liquor lees, and control medium)" and "(media containing CSL)" have been added (lines 100-102).

5) The sentence "The purchased mycelia of L. sordida arrived in a slant medium." have been added (line 108).

6) The description "were measured their weight." has been replaced to "were weighed" (line 122).

7) The sentence "All analyses were performed using R software (https://www.r-project.org/)" has been added (line 144).

8) Thank you for your valuable information. The reference has been cited and the sentences "There is a report that concentration of CSL changes production amount of ethanol by yeast and candida shehatae [24]. This previous report and the result in this study indi-cated that low-cost CSL is much better than control (YG)." (lines 152-154) and Reference No. 24 (lines 404-406) have been added.

9-12) The descriptions you pointed out in the comments only explain the experimental results in figures. In addition, we are unable to add proper references, because our discovery is new. We thought our English was poor and misleading, thus. the English has been improved by a professional service company.

13) The conclusion has been rewritten according to your suggestions (lines 307-313) and reference No. 43(lines 454-455) has been added.

The other parts we revised are as follows.

  1. i) The description of ", it was clarified that the optimum concentrations of CSL is 6% or less." has been replaced to "These results indicated that the optimal concentration of CSL is 6% or less." (lines 180-181).

I believe that this manuscript has been improved satisfactorily and is acceptable for publication in Journal of Fungi.

Sincerely yours,

Hajime Kobori, Ph. D.

Iwade Research Institute of Mycology Co., Ltd, Japan

Reviewer 2 Report

The paper reports on the production of Fairy chemicals by addition of corn steep liquor to liquid mycelium cultures of Lepista sordida. The paper is well written, interesting, and the conclusions are supported by the results. I have only very minor comments:

- methods: how many different liquid cultures were grown for independant culture media compositions?

- discussion: to produce a meaningful amount of fairy chemicals for an applications, how much liquid culture would be needed? Can the authors make some estimation about this?

Author Response

Dear Reviewer #2:

Thank you for your review about our manuscript entitled “Utilization of corn steep liquor for production of fairy chemicals by Lepista sordida mycelia”. We answered your comments as follows.

Your comments

The paper reports on the production of Fairy chemicals by addition of corn steep liquor to liquid mycelium cultures of Lepista sordida. The paper is well written, interesting, and the conclusions are supported by the results. I have only very minor comments:

1) methods: how many different liquid cultures were grown for independant culture media compositions?

2) discussion: to produce a meaningful amount of fairy chemicals for an applications, how much liquid culture would be needed? Can the authors make some estimation about this?

Our response to your comments

1) The sentence “Nineteen media were used in the experiments; 1) CSL, 2) Rice bran, 3) Wheat bran, 4) Japanese liquor lees, 5) CSL-1% (Salts-0.1+0.1% Glc-4% CSL-1%), 6) CSL-2%, 7) CSL-3%, 8) CSL-6% (Salts-0.1+0.1% Glc-4% CSL-6%), 9) Salts-0.1+0.1% Glc-2% CSL-1%, 10) Salts-0.1+0.1% Glc-1% CSL-1%, 11) Salts-0% Glc-4% CSL-1%, 12) Salts-0% Glc-2% CSL-1%, 13) Salts-0% Glc-1% CSL-1%, 14) Salts-0.1+0.1% Glc-2% CSL-6%, 15) Salts-0.1+0.1% Glc-1% CSL-6%, 16) Salts-0% Glc-4% CSL-6%, 17) Salts-0% Glc-2% CSL-6%, 18) Salts-0% Glc-1% CSL-6% and 19) control (YG) ” has been added in 2.2. Compositions of the media, Materials and Methods (lines 92-99)

2) The volume of the liquid media we used in this experiment was all 100 ml/500 ml Erlenmeyer flask. However, a biological reaction does not work well simply being increased proportionally the volume of the liquid medium and raw materials compared with a chemical reaction. Therefore, this consideration is an issue for future work.

I believe that this manuscript has been improved satisfactorily and is acceptable for publication in Journal of Fungi.

Sincerely yours,

Hajime Kobori, Ph. D.

Iwade Research Institute of Mycology Co., Ltd, Japan